# Topological interface states – a possible path towards a Landau-level laser in the THz regime

**Mark Oliver Goerbig***

Laboratoire de Physique des Solides, Université Paris Saclay,
CNRS UMR 8502, F-91405 Orsay Cedex, France

★ mark-oliver.goerbig@universite-paris-saclay.fr

## Abstract

Volkov-Pankratov surface bands arise in smooth topological interfaces, *i.e.* interfaces between a topological and a trivial insulator, in addition to the chiral surface state imposed by the bulk-surface correspondence of topological materials. These two-dimensional bands become Landau-quantized if a magnetic field is applied perpendicular to the interface. I show that the energy scales, which are typically in the $10-100$ meV range, can be controlled both by the perpendicular magnetic field and the interface width. The latter can still be varied with the help of a magnetic-field component in the interface. The Landau levels of the different Volkov-Pankratov bands are optically coupled, and their arrangement may allow one to obtain population inversion by optical pumping. This could serve as the elementary brick of a multi-level laser based on Landau levels. Moreover, the photons are absorbed and emitted either parallel or perpendicular to the magnetic field, respectively in the Voigt and Faraday geometry, depending on the Volkov-Pankratov bands and Landau levels involved in the optical transitions.



# 1 Introduction

Landau levels (LLs), which arise due to the quantization of the electrons' energy in a strong magnetic field, have been regularly proposed to be a promising system for a frequency-tunable laser in the THz regime [1–5]. Indeed, upon a putative population inversion between the LLs $n$ and $n + 1$ in parabolic bands, one may expect cyclotron emission with a typical frequency $\Omega_{n+1,n} = \omega_c$ given by the cyclotron frequency $\omega_c = eB/m_B$, which is directly controlled by the strength of the magnetic field $B$ and the band mass $m_B$. In spite of this conceptually appealing proposal, the path to the realization of a working LL laser is barred by strong obstacles that are mainly concerned with population inversion. The latter requires rather long-lived electrons in the excited LL, but their lifetime is strongly reduced by non-radiative recombinations, namely Auger processes that are prominent due to the equidistant LL separation [6] (for a detailed discussion of these processes, see Ref. [7]). In such processes, an electron in the excited LL $n + 1$ can be promoted due to electron-electron interactions to the LL $n + 2$ while the required energy is provided by a simultaneous desexcitation of another electron from $n+1$ to $n$. Instead of using one excited electron to emit a photon of frequency $\omega_c$, two electrons in the LL $n + 1$ are thus lost without emission of any photon. Another obstacle equally related to equisitant LLs is reabsorption of cyclotron light due to the transition $(n + 1) \to (n + 2)$, which is resonant with the $(n + 1) \to n$ transition used in the emission of light [7–9].

Soon after the isolation of graphene, physicists explored this material in cyclotron-emission experiments in the perspective of realizing a LL laser [4, 5, 10]. Due to the linearly dispersing bands of graphene electrons in the vicinity of charge neutrality, the LL spectrum is given by $E_n = \pm \hbar(v/l_B)\sqrt{2n}$, in terms of the Fermi velocity $v \simeq 10^6$ m/s and the magnetic length $l_B = \sqrt{\hbar/eB} \simeq 26\,\text{nm}/\sqrt{B[\text{T}]}$, i.e. the levels are no longer equidistant. While the orders of magnitude with a fundamental gap of $\hbar\Omega_{1,0} \sim 100$ meV for magnetic fields $B \sim 10$ T are promising for possible THz applications, Auger processes remain a relevant source of non-radiative recombination processes also in these relativistic systems [11]. For example, while the $1 \to 0$ transition is no longer in resonance with the neighboring $2 \to 1$ transition, it is in resonance with the transition $4 \to 1$ due to the square-root dependence of the LLs on the level index $n$ [7, 12]. Furthermore, it has been shown that the optical phonon responsible for the G band in graphene (at $\sim 200$ meV) also enhances decay processes that are detrimental to population inversion [13]. The drawback of resonant transitions and enhanced Auger processes can to some extent be healed, e.g. in gapless HgCdTe, where the low-energy electrons are described in terms of so-called Kane fermions. While their zero-field spectrum is similar to that of massless Dirac fermions, LLs with even indices do not exist in the spectrum so that some transitions are absent, such as the above-mentioned transition $4 \to 1$ [7].

An extremely interesting route towards the realization of a LL laser is the use of Dirac materials with a (mass) gap $\Delta$ that is on the same order of magnitude as the typical LL spacing, i.e. in the 100 meV range, for systems with a characteristic velocity parameter of $v \simeq 10^6$ m/s. In this case, the LL spectrum is given by

$$E_{\lambda,n} = \lambda\sqrt{\Delta^2 + 2\hbar^2 v^2 n/l_B^2}, \tag{1}$$

where $\lambda = \pm$ is the band index. Indeed, if $\Delta \sim \hbar v/l_B$, the LL spectrum is neither (approximately) linear in $n$ and $B$ as it would be in the limit $\Delta \gg \hbar v/l_B$ nor does it follow the square-root dependence of graphene in the opposite limit $\Delta \ll \hbar v/l_B$. In this case, the absence of simultaneous resonant transitions suppresses both reabsorption and non-radiative Auger scattering. First encouraging results in this direction have been obtained in gapped HgTe/CdTe quantum wells [9].

Another system in which massive Dirac fermions occur is the interface of a topological and a trivial insulator, in the form of Volkov-Pankratov (VP) states [14–16]. The bulk-surface

correspondence for topological materials enforces indeed the occurence of a massless chiral state at such an interface. However, it has been shown that the interface spectrum is much richer in systems with smooth interfaces, *e.g.* when the gap changes over a certain distance $\ell$ that characterizes the interface width and that is larger than an intrinsic length $\lambda_C = \hbar v/\Delta$. In smooth interfaces between a topological and a trivial insulator, one finds a whole family of surface states the spectrum of which is given by [16]

$$\epsilon_m(\mathbf{q}) \simeq \lambda \hbar v \sqrt{\mathbf{q}^2 + 2m/l_S^2}. \tag{2}$$

Here, $\mathbf{q} = (q_x, q_y)$ is the two-dimensional (2D) wave vector in the interface, $m$ denotes the index of the surface band, and $l_S = \sqrt{\ell \lambda_C}$ is a characteristic length determining the extension of the interface states in the $z$ direction perpendicular to the interface. Equation (2) is indeed valid as long as the energy of the surface bands at $\mathbf{q} = 0$ is smaller than the bulk gap, $\sqrt{2m}\hbar v/l_S \leq \Delta$. The latter condition is equivalent to requiring that the interface width $\ell$ be larger than $m$ times the intrinsic length $\lambda_C$ [16]. The $m = 0$ surface state is precisely the chiral state that survives in the abrupt limit, $\ell \to 0$, while the VP states (for $m \neq 0$) disappear in the continuum of the bulk states as soon as $\ell < \lambda_C$. Notice that the formation of VP states is a universal property of topological materials that has been studied not only in topological insulators [16–21], but also in Weyl semimetals [22, 23], graphene [24], and topological superconductors [25].

Very recently, inter-VP transitions have been measured within magneto-optical spectroscopy in $Pb_{1-x}Sn_xSe$ crystals [26] in which the Sn concentration determines whether the system is a trivial or a topological (crystalline) insulator [27–29]. Moreover, the concentration determines the size of the bulk gap so that smooth interfaces may be obtained by molecular-beam epitaxy (MBE) in which the Sn concentration is smoothly varied during the growth process, and where the absolute band gap in the topological regime can be designed such as to be identical to that in the trivial insulator [26]. This allows for a strong versatility in the fabrication of interfaces of various widths and thus of systems with specially designed fundamental gaps

$$\Delta_{\mathrm{VP}} = \sqrt{2}\hbar v/l_S = \sqrt{2}\Delta\sqrt{\frac{\lambda_C}{\ell}} = \sqrt{\frac{2\hbar v\Delta}{\ell}}, \tag{3}$$

between the $m = 1$ VP and the chiral ($m = 0$) surface states.

In the present paper, I argue that smooth topological interfaces, such as in the above-mentioned $Pb_{1-x}Sn_xSe$ crystals, may be extremely promising systems for the realization of long-lived population inversion if a magnetic field is applied perpendicular to the interface that quantizes the 2D electronic motion in the interface into LLs. The main reason for this expectation is the fact that VP bands provide us with several families of LLs that can to some extent be brought into close energetic proximity with LLs of the chiral surface band. This would allow for devices similar to three- or four-level lasers in which population inversion could be more easily achieved than in the usual LL setup.

Furthermore, optical pumping and radiative desexcitation can be chosen to happen in different directions via an intelligent choice of the involved transitions. Indeed, while the optical selection rules in the Faraday geometry impose that the emitted or absorbed photons propagate in the direction of the magnetic field for a transition coupling the LLs $n$ and $n \pm 1$, it has been shown previously [30] that such transitions must obey an optical selection rule $m \to m$ for the VP states. The selection rules are inverted in the Voigt geometry, where the emitted or absorbed photon propagates in a direction perpendicular to the magnetic field. The underlying reason for these selection rules and their geometry dependence is an intriguing analogy between the spatially changing gap parameter and LL quantization. As shown in Ref. [30] and briefly reviewed below, the spatially varying gap parameter can be viewed as a

fake magnetic field that is oriented in the plane of the interface, and the characteristic length $l_S$ plays the role of an effective magnetic length. Via an intelligent choice of the geometry of a cavity hosting the topological material and the involved transitions, one may therefore expect to obtain a strong cyclotron emission in the direction of the interface while pumping the system with photons propagating perpendicular to the interface, or *vice versa*.

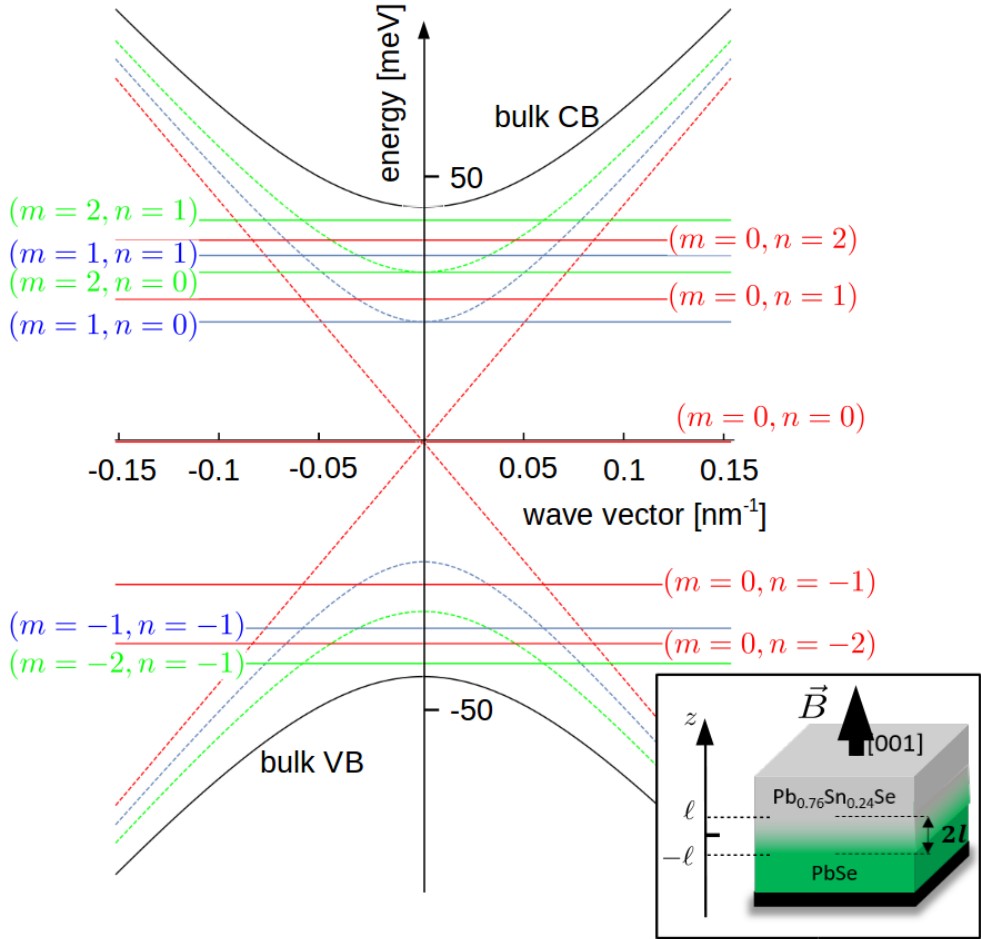

Figure 1: VP surface states. The surface bands for $m = 0$ (red) $m = \pm 1$ (blue) and $m = \pm 2$ (green) in the absence of a magnetic field are represented by dashed lines as a function of the wave vector. The black curves correspond to the bulk conduction (CB) and valence (VB) bands, for a typical value of $\Delta = 45$ meV. The interface width has been chosen to be $\ell = 100$ nm, here. The sign indicates the band index $\lambda$ for notational convenience. The LLs for a magnetic field of 1 T are shown as continuous lines with the colours corresponding to the surface bands. As a consequence of the parity anomaly, the $n = 0$ LL is found only in the upper VP band ($m = +1$). The inset, adapted from Ref. [26], shows the setup of a smooth interface between a trivial insulator (PbSe) at $z < -\ell$ and a topological insulator ($\text{Pb}_{0.76}\text{Sn}_{0.24}\text{Se}$) at $z > \ell$. The magnetic field is oriented in the [001] direction perpendicular to the interface.

## 2 Volkov-Pankratov states and optical selection rules in a magnetic field

Let us first review some basic features of VP states and their coupling to light in the presence of a magnetic field along the lines of Ref. [30]. We consider an interface between a trivial insulator in the lower part of the device ($z < -\ell$) and a topological one in the upper part ($z > \ell$) (see inset of Fig. 1). Such a situation may be modeled, *e.g.*, in terms of the following generic Hamiltonian [31],

$$H_0 = \Delta(z)\tau_z + \hbar v \left[ q_z \tau_y + \tau_x (q_y \sigma_x - q_x \sigma_y) \right], \tag{4}$$

where the Pauli matrices $\tau_\mu$ and $\sigma_\nu$ represent an orbital and the spin degree of freedom, respectively. Quite generally, the Hamiltonian describes a massive Dirac fermion the mass of which vanishes for $\Delta(z_0) = 0$ at the position $z_0$ that we choose to be zero and that characterizes the topological phase transition. Exchanging $\tau_z$ and $\tau_x$ via the global unitary transformation $T = \exp(i\pi\tau_y/4)$ – a rotation by $\pi/2$ around the $y$-axis in orbital space – allows one to rewrite the Hamiltonian in the form

$$\tilde{H}_0 = \begin{pmatrix} \hbar v(q_y \sigma_x - q_x \sigma_y) & [-\Delta(z) - i\hbar v q_z]\mathbb{1} \\ [-\Delta(z) + i\hbar v q_z]\mathbb{1} & -\hbar v(q_y \sigma_x - q_x \sigma_y) \end{pmatrix}, \tag{5}$$

where $\mathbb{1}$ represents the identity in spin space. This form unveils the analogy between the quantization of the electronic motion in the $z$-direction and LL quantization. Indeed, if we linearize the gap inversion over the smooth interface by a linear function connecting a gap parameter of $+\Delta$ in the trivial insulator (at $z < -\ell$) and $-\Delta$ in the topological insulator (at $z > \ell$), *i.e.* $-\Delta z/\ell$, the system may be mapped to the LL problem of massive Dirac fermions [16],

$$\tilde{H}_0 = \hbar v \begin{pmatrix} q_y \sigma_x - q_x \sigma_y & \frac{\sqrt{2}}{l_S} c\mathbb{1} \\ \frac{\sqrt{2}}{l_S} c^\dagger\mathbb{1} & -q_y \sigma_x + q_x \sigma_y \end{pmatrix}, \tag{6}$$

in terms of the harmonic-oscillator ladder operators

$$c = -\frac{1}{\sqrt{2}}\left(\frac{z}{l_S} + iq_z l_S\right), \quad \text{and} \quad c^\dagger = -\frac{1}{\sqrt{2}}\left(\frac{z}{l_S} - iq_z l_S\right), \tag{7}$$

where $l_S = \sqrt{\ell\hbar v/\Delta}$ is the above-mentioned length characterizing the wave-function extension in the $z$ direction. It plays the role of a "fake" magnetic length. Indeed, within this analogy, the variation of the gap parameter in the $z$ direction may be viewed as a vector potential that generates a "fake" magnetic field oriented in the interface, while the physical magnetic field is oriented in the $z$ direction. In the absence of the latter, the diagonalization of Hamiltonian (6) yields the spectrum (2) of the VP surface bands. Notice furthermore that the above description can easily be generalized to a situation where the gap in the topological insulator is not of the same size as that in the trivial one [16], in which case the effective interface width $l_S$ is determined by an average between the two gaps.

When a magnetic field is applied to the system in the $z$-direction, the VP bands (2) get quantized into LLs whose spectrum reads (for $n \neq 0$)

$$E_{\lambda,m,n\neq 0} = \lambda\hbar v\sqrt{\frac{2|m|}{l_S^2} + \frac{2|n|}{l_B^2}}. \tag{8}$$

For notational simplicity, we merge the band index $\lambda$ from now on with the VP and LL indices so that $(-m, -n)$ corresponds to the $n$-th LL in the $m$-th VP band of negative energy ($\lambda = -$),

whence the modulus of the indices in the spectrum (8) to avoid confusion. Due to the parity anomaly, the above spectrum is only valid for LLs with an index $n \neq 0$, while the $n = 0$ LLs of the VP bands stick either to the bottom of the positive-energy bands ($\xi = +$) or to the top of the negative-energy VP state ($\xi = -$)

$$E_{m,n=0} = \xi \hbar v \sqrt{\frac{2|m|}{l_S^2}}, \tag{9}$$

depending on the chirality index $\xi$. The latter can be changed if we change the order between the topological and the trivial insulator (interface between a topological insulator in the lower part and a trivial one in the upper part), and it can also be altered easily by changing the orientation of the magnetic field. The LL spectrum for the $m = 0, \pm 1$ and the $\pm 2$ VP states are shown in Fig. 1, for typical values of the relevant energy and length scales in PbSnSe crystals (see Sec. 5).

Notice that the surface-state-width parameter $l_S$ can be decreased effectively with the help of an *inplane* magnetic field $B_\parallel$. In order to appreciate this effect, let us consider the Landau gauge $\mathbf{A} = (0, -B_\parallel z, 0)$ which conspires with the gap variation $\Delta(z) = -\Delta z / \ell$ in that it does not affect the (free) electronic motion in the $xy$-plane. In this case, the Hamiltonian may still be brought into the form (6), albeit with a more complicated unitary transformation $T = \exp(i\theta \tau_y \sigma_x / 2)$ that also acts on the spin sector [30]. Here, the angle $\theta$ is given by the ratio $\theta = \arccos(l_S^2 / l_S(B_\parallel = 0)^2)$ between the enhanced width $l_S$ and the original width parameter $l_S(B_\parallel = 0) = \sqrt{\ell \lambda_C}$ in the absence of an inplane magnetic field. Their relation is given by the geometric form $l_S^{-4} = l_S(B_\parallel = 0)^{-4} + (eB_\parallel / \hbar)^2$ so that the effective energy separation between the VP states, given by Eq. (3) is increased to

$$\Delta_{\mathrm{VP}} = \sqrt{\frac{2\hbar v \Delta}{\ell}} \left(1 + \frac{e^2 v^2 B_\parallel^2 \ell^2}{\Delta^2}\right)^{1/4}. \tag{10}$$

The analogy between interface width and magnetic field finally yields a physical understanding of the optical selection rules between the levels $(m, n)$ and $(m', n')$, where I recall that the first index indicates the VP band and the second one the physical LL. In the Faraday geometry, in which the absorbed or emitted photon has a wave vector and thus propagates in the direction of the magnetic field,[1] angular-momentum conservation imposes that the only optically active transitions involve adjacent LL indices, $n \to n' = \pm(n \pm 1)$, regardless of the band index $\lambda$. The relative sign in the bracket on the right-hand side of the selection rule is determined by the circular polarization of the photon, $\sigma^-$ coupling to the transition $n \to \pm(n + 1)$ and $\sigma^+$ to $n \to \pm(n - 1)$. This needs to be contrasted to the Voigt geometry, where the photon propagates in the plane perpendicular to the magnetic field and where the LL index remains unchanged $n \to n' = \pm n$ if we consider the electric field $\mathcal{E}$ of the photon to be oriented in the same direction as the external magnetic field. Since the fake magnetic field that yields the VP bands is oriented in the interface, Voigt and Faraday geometry are inverted, and a photon propagating perpendicular to the interface couples VP bands with the same index ($m \to m' = \pm m$) while a photon with a wave vector in the interface couples adjacent VP bands [$m \to m' = \pm(m \pm 1)$]. As in the LL problem, the selection rules, which are summarized in the table above, do not depend on the band index. In both cases, VP states and LLs, it is the circular polarization of the photon that determines which of the adjacent levels or bands are optically coupled.

---

[1] I define the Faraday and Voigt geometry with respect to the direction of propagation, which due to Maxwell's equations (in the absence of free charges), $\nabla \cdot \mathcal{B} = 0$ and $\nabla \cdot \mathcal{E} = 0$ is perpendicular to the polarization plane spanned by the field-components $\mathcal{E}$ and $\mathcal{B}$ of the light field. In the Faraday geometry, the external magnetic field is therefore perpendicular to the polarization plane, while it is oriented in the polarization plane in the Voigt geometry. In the latter, we need in addition to specify the direction of the $\mathcal{E}$-field, which is parallel to the external magnetic field.

Table 1: Optical selection rules in the Faraday (photon propagation perpendicular to the interface) and the Voigt geometry (photon propagation in the interface with the electric-field component $\mathcal{E}$ of the photon parallel to the direction of the external magnetic field).

| geometry | VP states | Landau levels |
|:---:|:---:|:---:|
| Faraday | $m \to \pm m$ | $\pm n \to n \pm 1$ |
| Voigt ($\mathcal{E} \parallel B$) | $m \to \pm m \pm 1$ | $n \to \pm n$ |

## 3 Three-level scheme

Let us first illustrate schematically the different emission processes in terms of resonant (optical) pumping within a three-level picture to fix some basic ideas. In a first step, we consider the situation depicted in Fig. 2(a) where the LL energy scale $\sqrt{2}\hbar v/l_B$ is slightly larger than the VP gap $\Delta_{\mathrm{VP}}$ given in Eq. (3), *i.e.* the magnetic length is slightly smaller than the effective surface width $l_S$. We show below in Sec. 5 that this situation can be easily achieved *e.g.* in MBE-grown $Pb_{1-x}Sn_xSe$ crystals. In this case, the $n = 1$ LL of the chiral $m = 0$ surface state is slightly above the $n = 0$ level of the upper VP band with an index $m = 1$. In Fig. 2(a), we consider optical pumping in the Faraday geometry, where the light frequency is resonant with the $(m = 0, n = 0) \to (m = 0, n = 1)$ transition. If the target level is only slightly above the lowest LL of the $m = 1$ VP band, $(m = 1, n = 0)$, one may expect rapid non-radiative decay of the excited electrons to the latter level. These electrons may then decay to the zero-energy level $(m = 0, n = 0)$ by emitting light with a frequency $\omega = \sqrt{2}v/l_S$ in the Voigt geometry, *i.e.* absorbed and emitted photons, even if they may be almost resonant, propagate in perpendicular directions.

While the magnetic field does then not allow one to control the frequency of the transition, which is determined by $l_S$ and thus by the interface width $\ell$, it allows us to bring the levels $(m = 1, n = 0)$ and $(m = 0, n = 1)$ into close energetic vicinity and thus to increase the transition rate between the two levels, which is proportional to

$$\Gamma \sim \frac{1/\tau}{1/\tau^2 + 2v^2(1/l_B - 1/l_S)^2},\tag{11}$$

if we consider Lorentzian level broadening due to a dephasing time $\tau$ [32]. For a typical value of $\tau \sim 100$ fs, the level broadening is then on the order of some meV. Notice, however, that the frequency of the emitted light may to some extent be varied with the help of an inplane magnetic field, according to Eq. (10).

Similarly, one may use the Voigt geometry for pumping the transition $(m = 0, n = 0) \to (m = 1, n = 0)$. If the latter is now slightly above the $(m = 0, n = 1)$ level [see Fig. 2(b)], *i.e.* for smaller magnetic fields with $\sqrt{2}\hbar v/l_B < \Delta_{\mathrm{VP}}$, the $n = 1$ LL of the chiral surface band may be populated by non-radiative decay processes, and one may expect a population inversion between the $(m = 0, n = 0)$ and $(m = 0, n = 1)$ levels, with cyclotron emission at the fundamental frequency $\omega_C = \sqrt{2}v/l_B$. As before, the emitted photon propagates then in a direction perpendicular to that of the absorbed photon, but Faraday and Voigt geometries are inverted.

## 4 Four-level scheme

We now investigate a possible four-level scheme for population inversion, as shown in Fig. 3. For the sake of the argument, we consider the $n = 0$ LLs of the VP bands now to be situated in

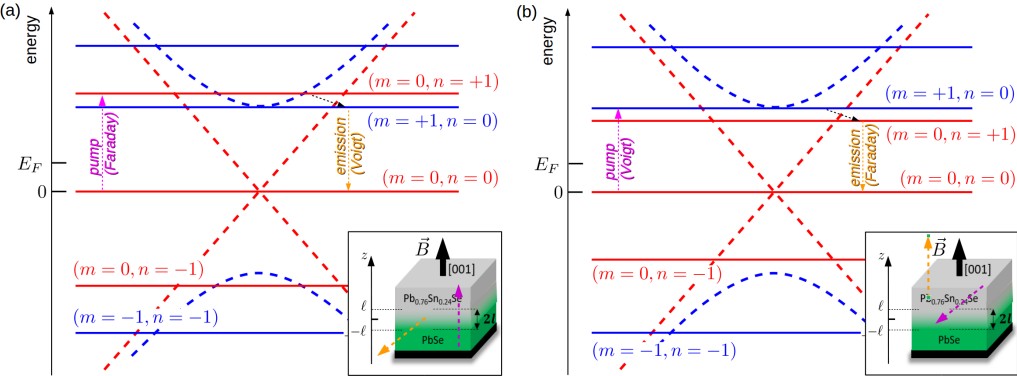

Figure 2: Sketch of a three-level scheme for stimulated cyclotron emission. As in Fig. 1, the dashed lines indicate the dispersion of the surface bands in the absence of a magnetic field while the full lines represent the associated LLs. Ideally, the Fermi level ($E_F$) is considered to be situated above the ($m = 0, n = 0$) level so that the zero-energy level is filled. (a) Pumping in the Faraday geometry for $\sqrt{2}\hbar v/l_B > \Delta_{\mathrm{VP}}$. The ($m = 0, n = 1$) level is populated via pumping (dashed magenta arrow) from the zero-energy ($m = 0, n = 0$) level, and emission can take place in the Voigt geometry in the transition ($m = 1, n = 0$) $\rightarrow$ ($m = 0, n = 0$) (dashed orange arrow). The level ($m = 1, n = 0$) is rapidly populated by the ($m = 0, n = 1$) level if the latter is almost resonant with the former, via a rapid non-radiative decay (dashed black arrow). (b) Similar process with pumping in the Voigt geometry for $\sqrt{2}\hbar v/l_B < \Delta_{\mathrm{VP}}$. The pumping transition is ($m = 0, n = 0$) $\rightarrow$ ($m = 1, n = 0$) (dashed magenta arrow) while emission takes place in the ($m = 0, n = 1$) transition (dashed orange arrow) that is populated via rapid non-radiative decay processes (dashed black arrow) from the ($m = 1, n = 0$) level, which is slightly higher in energy. The insets represent the device geometries, with the direction of propagation of the absorbed (dashed magenta arrow) and emitted (dashed orange arrow) photons, for the two configurations, respectively.

the negative-energy branch. As already mentioned, this can easily be achieved by switching the orientation of the magnetic field. Let us choose optical pumping by light in the Voigt geometry that is resonant with the transition ($m = 0, n = -1$) $\rightarrow$ ($m = 1, n = 1$). In contrast to the three-level scheme discussed in the previous section, the target level is no longer in close vicinity with the level below that is ($m = 0, n = 1$). However, both are optically coupled, and an electron can transit from ($m = 1, n = 1$) to ($m = 0, n = 1$) by emitting a photon in the Voigt geometry again. While this photon is sacrificed in the present scheme, its emission allows for an enhanced population of the $n = 1$ LL in the chiral surface band. This is particularly interesting since the transition ($m = 0, n = 1$) $\rightarrow$ ($m = 0, n = 0$) to the central zero-energy level, which we consider to be non or only sparsely populated, is resonant with the ($m = 0, n = 0$) $\rightarrow$ ($m = 0, n = -1$) transition to the original level served in the pumping process. Under strong pumping and thus a strong depletion of the ($m = 0, n = -1$) level, it is therefore possible to emit *two* photons at the cyclotron frequency.

It is noteworthy that the above-mentioned resonant cyclotron transitions are also involved in non-radiative Auger processes. Such Auger processes have been shown to be detrimental to population inversion in GaAs and graphene [6, 7, 11]. Here, however, this is not the case. Indeed, one of the two electrons that take part in the Auger process, where both electrons originally reside in the ($m = 0, n = 0$) LL, is kicked back into the ($m = 0, n = 1$) LL thus maintaining the fertile population inversion. While the electron that transits simultaneously

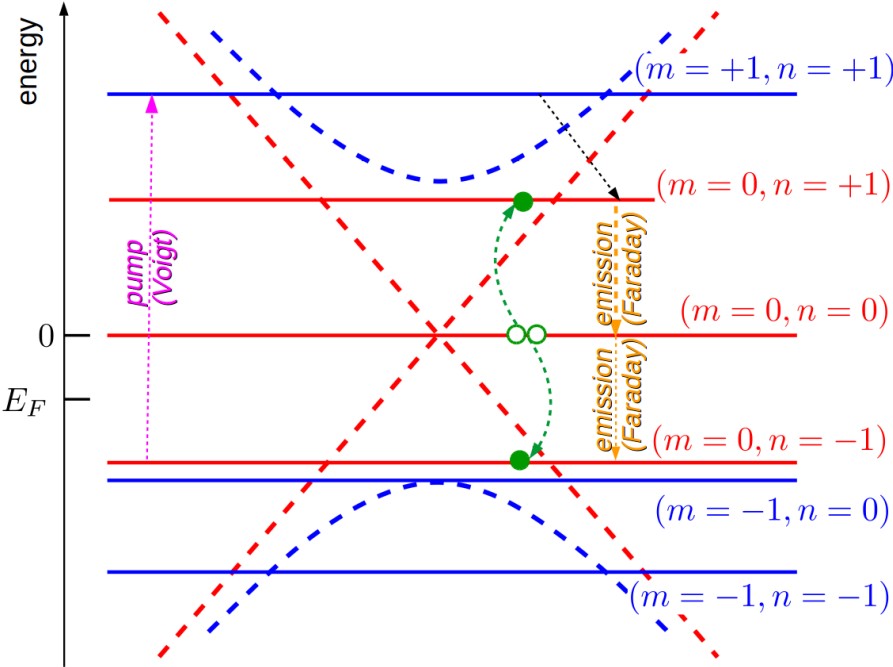

Figure 3: Sketch of a four-level scheme for stimulated cyclotron emission. As in Fig. 1, the dashed lines indicate the dispersion of the surface bands in the absence of a magnetic field while the full lines represent the associated LLs. The Fermi level ($E_F$) is now situated below the ($m = 0, n = 0$) level so that the zero-energy level is empty. We consider pumping in the Voigt geometry for $\sqrt{2}\hbar v / l_B < \Delta_{\text{VP}}$ and a situation, where the $n = 0$ LL of the $m = 1$ VP state is in the negative-energy branch. The ($m = 1, n = 1$) level is populated via pumping (dashed magenta arrow) from the ($m = 0, n = -1$) level, and the pumped electrons can decay radiatively to the ($m = 0, n = 1$) level by emitting light also in the Voigt geometry. Electrons in the ($m = 0, n = 1$) LL can then decay to the zero-energy level ($m = 0, n = 0$) emitting cyclotron light in the direction perpendicular to the interface (in the Faraday geometry, dashed orange arrows). Furthermore, this transition is resonant with ($m = 0, n = 0$) → ($m = 0, n = -1$), and a second photon with the cyclotron frequency can thus be emitted. Auger processes, shown in green now enhance the depopulation of the zero-energy LL ($m = 0, n = 0$) and are therefore helpful for cyclotron emission.

to the level ($m = 0, n = -1$) is energetically lost, *i.e.* it does not emit a photon, the first electron emits another photon at the cyclotron frequency before it can take part in another Auger process or radiatively transit to ($m = 0, n = -1$).

# 5 Possible realization in Pb$_{1-x}$Sn$_x$Se crystals

While the above arguments are not restricted to a particular topological insulator, it is useful to discuss the orders of magnitude of the probably best controlled system in which VP states occur that is MBE-grown Pb$_{1-x}$Sn$_x$Se crystals [27–29]. As mentioned in the introduction, the MBE growth allows one to obtain interfaces with a well-controlled interface width in which the VP states obey to great accuracy the dispersion (2) [26]. Most saliently, the Sn concentration $x$ in the Pb substitution allows one to trigger the electronic nature of the material. While, for $x = 0$, the system is a trivial band insulator it becomes a crystalline topological insulator

above a critical concentration on the order of ($x_c \simeq 0.12$). Moreover, the Sn concentration determines the size of the gap, which is on the order of 90 meV in the trivial insulator at $x = 0$ [28, 29]. The choice $x = 0.24$ allows one to obtain the same magnitude for the gap in the topological insulator ($2\Delta \sim 90$ meV) [26], but even larger gaps on the order of $2\Delta \sim 200$ meV may be obtained upon variation of temperature and strain on the crystals [27, 29].

Magneto-optical experiments indicate that the fundamental VP gap (3) scales as [26]

$$2\Delta_{\mathrm{VP}} \simeq 45\,\mathrm{meV}/\sqrt{\ell/100\,\mathrm{nm}}\,, \tag{12}$$

and samples with interface widths between $\ell = 50$ and 200 nm have been obtained, while the intrinsic length has been estimated to be $\lambda_C \simeq 6$ nm so that the effective surface width varies between $l_S \sim 17$ nm and $l_S \sim 35$ nm. In order for the magnetic length to be on the same order of magnitude as $l_S$ – situation considered in the present paper – one would require magnetic fields in the range 0.5...3 T that are easily accessible experimentally.

Notice that the Fermi velocity is roughly half of that in graphene so that $v/c \sim 1/600$, in terms of the speed of light $c$. If we consider the fundamental cyclotron resonance associated with the transition ($m = 0, n = 1$) → ($m = 0, n = 0$), the energy of the transition is thus roughly

$$\sqrt{2}\frac{\hbar v}{l_B} \simeq 20\,\mathrm{meV} \times \sqrt{B[\mathrm{T}]}\,. \tag{13}$$

This implies a the transition rate [4]

$$\Gamma_{(m=0,n=1)\to(m=0,n=0)} = 2\alpha \left(\frac{v}{c}\right)^2 \omega \tag{14}$$
$$\simeq 2.4 \times 10^6\,\mathrm{s}^{-1} \times \sqrt{B[\mathrm{T}]}\,,$$

in terms of the fine-structure constant $\alpha = 1/137$, if we consider dipolar light coupling. This is roughly a factor of four smaller than in graphene due to the reduced Fermi velocity $v$. Notice that interaction-induced decay processes take place at much shorter time scales, typically in the fs range. In the case of almost resonant levels, as discussed in the previous section [see e.g. the levels ($m = 1, n = 0$) and ($m = 0, n = 1$) in Fig. 2(a) and (b)], Fermi's golden rule indicates that the decay rate from the higher to the lower level is on the order of [32]

$$\Gamma \sim \frac{2\pi}{\hbar} |\langle m = 1, n = 0 |V_C| m = 0, n = 1\rangle|^2 f(\omega, \Delta E, \tau)$$
$$\sim \frac{2\pi}{\hbar} \left(\frac{e^2}{\epsilon l_B}\right)^2 \left(\frac{\tau}{\hbar}\right) \simeq \epsilon^{-2} \times 10^{16}\,\mathrm{s}^{-1} \times B[\mathrm{T}]\,, \tag{15}$$

where $\epsilon$ is the dielectric constant of the host material. Here, $\langle m=1, n=0|V_C|m=0, n=1\rangle \sim e^2/\epsilon l_B$ is, apart from a possible numerical prefactor, the Coulomb-interaction matrix element between the states associated with the two almost resonant levels (of energy difference $\Delta E$), and

$$f(\omega, \Delta E, \tau) = \frac{\hbar/\tau}{(\Delta E - \hbar\omega)^2 + \hbar^2/\tau^2}\,, \tag{16}$$

is the distribution, which may approximated by a Lorentzian with a typical width given by the dephasing time $\tau$ and that reduces to a Dirac peak in the limit $\tau \to \infty$.

One should also notice that the ($m = 1, n = 1$) level, used e.g. in the above four-level scheme, can also be brought into close energetic vicinity of the bottom of the *bulk* conduction band at $\Delta \sim 45...100$ meV. While the schemes discussed above in terms of resonant pumping in a specific geometry require themselves a THz source, one might then alternatively use the bulk conduction band as a target of pulsed or continuous pumping at higher energies and

rely on rapid decay processes towards the band bottom, which then serves as a reservoir for the $(m = 1, n = 1)$ level. However, to test this possibility, one would need to rely on a decay towards this target level at the interface that is more rapid than the bulk recombination. Being aware of the relevance of a detailed balance equation for the involved transition rates and quantitative evaluation of the latter, such a quantitative study is nevertheless beyond the scope of the present paper and left for future investigations.

Let us finally mention the role of the topological-insulator surface opposite to the active interface investigated here. While the latter is grown such as to provide a smooth interface, the former is considered as sharp so that it does not host VP states. However, it necessarily hosts a chiral surface state that, in the presence of the perpendicular magnetic field, might in principle absorb photons via LL transitions $(m = 0, n) \rightarrow (m = 0, \pm n \pm 1)$ that have previously been generated in the active interface within the Faraday geometry. In order to avoid such unwanted reabsorption, one needs to passivate the surface by a judicious choice of the Fermi level. This can, *e.g.*, be achieved by a gate or probably more realistically by chemical electron doping with the help of adatoms so that an absorption to the level $(m = 0, n = 0)$ becomes Pauli-blocked in the schemes discussed above.

## 6 Conclusions

In conclusion, I have argued that the particular surface-state spectrum, which is formed in smooth interfaces between a trivial and a topological insulator, is a promising path towards the LL laser. A tunable LL laser would indeed be of relevance in the quest for a coherent-light source in the THz range, where the lack of such source is known as the "terahertz gap" [33, 34]. In addition to the chiral surface state, which may be described in terms of a massless Dirac fermion, VP states are formed if the gap parameter varies over a width $\ell$ that must be larger than the intrinsic length scale $\lambda_C = \hbar v / \Delta$, in terms of the bulk gap $\Delta$. These surface bands have the form of a massive 2D Dirac fermion, and each of the bands gives rise to LLs if a magnetic field is applied perpendicular to the interface. One is thus confronted with families of LLs the energy of which can to great extent be controlled by the magnetic field for the LL separation and by the interface width for the energy separation between the VP bands. While the latter is given by the sample growth, it can still be varied *in situ* with the help of an inplane magnetic field that effectively reduces the interface width and thus increases the gap between the VP bands.

The magnetic field does not only allow one to change the cyclotron frequency, at which light is emitted in certain setups, but also to bring LLs associated with different VP bands into close energetic vicinity. When the gap between the VP bands is on the same order of magnitude as the typical LL separation – this situation can be easily achieved experimentally, *e.g.* in $Pb_{(1-x)}Sn_x Se$ crystals – the LL spectra are neither equidistant nor follow a square-root law so that both Auger and reabsorption processes are maximally suppressed. Futhermore, remaining Auger processes involving the central $(m = 0, n = 0)$ level and the adjacent target levels $(m = 0, n = \pm 1)$ may even be helful to maintain population inversion.

Another highly unusual and, for devices, potentially extremely fertile aspect of light emission in VP LLs is the direction of propagation of the absorbed and emitted photons. Indeed, photons with a wave vector perpendicular to the interface (Faraday geometry) are absorbed and emitted in transitions involving adjacent LL indices $n$ and $n \pm 1$ but the same VP band index $m$, regardless of whether the LLs are formed in the positive- or negative-energy branch of the VP bands. On the other hand, photons propagate inside the interface (Voigt geometry) for transitions $(m, n) \rightarrow (m \pm 1, n)$. This would allow for a smart design of the Fabry-Pérot cavities such that the extension in the $z$- and $x/y$-directions match the photon wavelength of the respective transitions, especially if pumping and emission are associated with the two different geometries (Faraday and Voigt).

# Acknowlegments

I would like to thank Gauthier Krizman, Louis-Anne de Vaulcher, and Milan Orlita for fruitful discussions.

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
