# Peer review of "Topological interface states -- a possible path towards a Landau-level laser in the THz regime"

_SciPost Physics, doi:SciPost Phys. 15, 246 (2023)_

## Round 1 · Referee Report · Anonymous · 2023-9-18

Strengths

Original, well written, well argued, at the crossroads of fundamental and applied physics.

Report

In this manuscript, the author proposes a new mechanism to engineer lasing in the THz regime. For that purpose, the author takes advantage of the Landau levels (LLs) made out of Volkov-Pankratov (VP) states in a magnetic field. Those VP states are massive states that appear at smooth interfaces between topologically distinct materials, in addition to the topologically protected massless surface state that survives for abrupt interfaces. Interestingly, the author notices that a population inversion of electrons, induced by resonant optical pumping, can be controlled by a combination of the magnetic field (controlling the LLs spacing) and the interface width (controlling the dispersion of the VP states). The comparison between the magnetic length and the surface-state-width parameter, that somehow plays the role of a fictitious magnetic length, may give rise to two different scenarios of photons emissions, called Faraday and Voigt configurations, in which the electrons relaxation or excitation mechanism implies different transitions between LLs of VP states and LLs of the topological chiral interface state. Besides, those two configurations interestingly correspond to photons emissions in either parallel or perpendicular direction to the external magnetic field. The author also investigates different schemes implying three and four LLs, and argues a possible implementation in a PbSnSe crystal whose concentration in Pb and Sn is known to control a gap inversion between a topologically trivial and a crystalline topological insulator.

Beyond finding this original proposal very appealing, I also find the paper very well written and argued. Many orders of magnitude are provided, an example of a concrete material is suggested, transition rates are discussed. I am confident that this convincing idea will stimulate further promising studies. I am thus very inclined to recommend this paper for publication.

I nevertheless would like to ask one (maybe naive) question to the author, about the sample geometry. All the physics discussed here takes place in the vicinity of a bulk interface in a 3D material (see e.g. insets in Fig. 1 and 2). When the photons are emitted, they are assumed here to propagate freely through the bulk and then escape out the material. In particular, when pumping in the Voigt geometry, a photon is emitted between LLs of the chiral surface state (Faraday emission). This exact same transition exists at the actual surface of the topological material (where no VP exist), and toward which the emitted photon is sent, so that this photon could be reabsorbed at the surface. Could this mechanism, not mentionned in the manuscript, spoil the required lasing effect, at least in that geometry? For instance, we could imagine that the photon could be re-emitted, but in another direction.

I list below a few (possible) typos.

Below equation 2, after [16], is that the n=0 surface state or the m=0 surface state?
Before Part II - one may therefore expect(ed)
Before Eq. 7. I guess \ell should be \ell_S
Below Eq. 7, after \tau, there is a missing reference ‘’[]’’
Beginning of the last paragraph of section III, ‘’goemetry ‘’ -> ‘’geometry’’
Last paragraph, first column, page 5, a reference should be added regarding after ‘’Such Auger processes have been shown…’’
Page 6, first paragraph, there is a latex problem with parenthesis Pb_{(1-x)}.

---

## Round 1 · Referee Report · Anonymous · 2023-9-19

Strengths

1. Draws interesting and promising connections between distinct fields of research
2. Good balance of review and new results
3. Provides intuitive understanding of the electronic states under examination

Weaknesses

1. It is essential that more detailed calculations of optical processes are included.
2. A stronger motivation of the actual advantages of this proposal in view of devices is needed.
3. Some points of the presentation should be clarified.

Report

The manuscript by Goerbig capitalizes on previous works (including the author's ref.16) on the electronic states at the interface between a topological and trivial 3D material to propose an interesting application of topological interfaces for optical devices. As such, the manuscript contains an interesting and commendable pioneering attempt to connect the normally very distinct worlds of abstract topological condensed matter theory and real-word optoelectronic applications. In a sense, this proposal generalizes what is often done in current semiconductor technology to engineer electronic states of, e.g., quantum wells (the extreme example is the quantum cascade laser) to a novel class of materials with richer possibilities stemming from new alloys with topological band structures. On this basis, I find that the manuscript has the potential to become a very good SciPost paper.

However, before I can recommend it for publication, the author should carefully consider the following list of remarks:

1/ I am very confused when the author speaks of Voigt vs. Faraday geometry and characterizes the optical process in terms of the direction of propagation. As far as I understand, the wavevector of light involved in the optical transition is very small and thus negligible, while what really matters in the selection rules for optical transitions is the direction of polarization of the electric field. The author should correspondingly correct the manuscript to make this point fully clear for the readers.

2/ After eq.(5) (and at a few other places in the ms.) the author says that an in-plane magnetic field may modify the surface-state-width parameter. I do not understand how this can happen, so I recommend that the author adds some explaination to make the manuscript self-contained.

2bis/ A few lines later, the author says 'Within this analogy... "fake" magnetic field oriented in the interface... in the z-direction'. Also this sentence is obscure. I suspect it is somehow connected to my point 2/: if so, the author should clarify.

3/ If I understand correctly, the proposal of Sec.III requires that the upper state of the population-inverted transition is only slightly below the optically pumped state. As such, I don't understand what would be the advantage of using this device as a source of light as compared to directly using the pump laser.

3bis/ Some justification to the parameter dependences shown in formula (31) would also be appreciated. It could be useful to mention this formula also around eq.(7).

4/ To make the manuscript complete, the author could include more levels in fig.2 so to clearly show that no other optical transition to higher levels is indeed possible. As further modification to the figure, he could explicitly indicate the Fermi level (right above 0, I guess) and use a logical color code for the arrows indicating pump and emitted light.

5/ In fig.3, he should also indicate the Fermi energy which is now just below 0. He should add more levels to show that no higher level is populated.

5bis/ As a most important point, the author should add some explicit discussion and/or numerical rate equation calculation that explicitly demonstrates that population inversion is indeed possible and no nasty reabsorption process can take place between the several levels involved in the optical transitions. The arguments on the RHS column of pag.5 (including the mentioned but not calculated depletion of the m=0,n=-1 level) are in fact too qualitative to be fully convincing for optics-oriented readers.

6/ In general, readers would appreciate knowing more on the potential advantage that the LL laser device would have in comparison to other existing or proposed sources of coherent light in the IR range.

7/ I do not generally like very long paragraphs in a research article. I suggest the author to add more \newline's to facilitate the reader in following the different steps of the logical flow.

Requested changes

1. Reinforce the discussion of the optical processes at play and, if possible, make it quantitative.

2. Clarify the role of light propagation direction vs. polarization in the analysis of the Faraday vs. Voigt geometry

3. Update the figures

4. Take all other remarks in due consideration

---

## Round 3 · Referee Report · Anonymous (Referee 2) · 2023-11-21

Report

I am (almost) satisfied by the author's revisions in response to my report. I can (almost) recommend publication provided the author clarifies a couple of last points (likely due to my ignorance, but still worth consideration):

1) I am still puzzled by the Faraday vs. Voigt geometry. I try to explain my doubt: the propagation direction does not fully determine the polarization direction (that is, the E field of the e.m. wave), but only imposes that E is orthogonal to k. In the Faraday geometry, this is not a problem, as any E orthogonal to k will be orthogonal to the static B0. In the Voigt geometry, there is a key difference between having E orthogonal to B0 and having E parallel to B0. I suspect that by Voigt geometry, the authors only refers to this latter case. If it is so, he should explicitly mention it in the ms. and, perhaps, in the insets of fig.2. If not, it is even more important that he clarifies the ms. to avoid other readers to fall into the same trap into which I have fallen. (An explicit discussion of the E-field polarization is all the way more important as emission processes are not limited to k parallel or orthogonal to B0)

1bis) For the sake of completeness, when discussing the Faraday geometry, the author may separately indicate the selection rules that apply under sigma+ and sigma- polarized light.

2) The author should explain in the captions of figs.2 and 3 what are the red and blue dashed lines. I suspect they are the dispersion of VP states in the absence of B, but he should specify it explicitly.

Requested changes

  1. Take my remarks into due consideration

---

## Round 3 · Author Response

Warnings issued while processing user-supplied markup:

  • Inconsistency: plain/Markdown and reStructuredText syntaxes are mixed. Markdown will be used.
    Add "#coerce:reST" or "#coerce:plain" as the first line of your text to force reStructuredText or no markup.
    You may also contact the helpdesk if the formatting is incorrect and you are unable to edit your text.

Dear Editor,

thank you very much for sending me the two detailed and globally very positive reports on my manuscript "Topological interface states -- a possible path towards a Landau-level laser in the THz regime". I have amended the manuscript according to the Referees' requests, and please find enclosed a reply to both Referees.

I hope that the manuscript is now deemed suitable for publication in SciPostPhys.

Sincerely yours, Mark Oliver Goerbig

Reply to "Anonymous Report 1"

I would like to thank the Referee for her/his detailed and positive report on my manuscript, especially for the indeed relevant question raised in the third paragraph.

The Referee is absolutely right that the opposite surface of the topological insulator, albeit sharp, has the capacity to reabsorb photons emitted in the $(m=0,n=1)->(0,0)$ or $(0,0)->(0,-1)$ transitions since it hosts a chiral surface state that is submitted to Landau-level quantization. In order to suppress such unwanted reabsorption processes, the most promising path is to Pauli-block this transition at the sharp interface. This might be achieved by positioning the Fermi level between the (0,0) and the (0,1) state, e.g. by an electric gate or, more promisingly, by chemical doping with electron donors. I have added a paragraph at the end of Sec. V in response to this point raised by the Referee.

I am also grateful for the minor points (typos) which the Referee pointed out and that have been corrected. Notice just that one may use either $\ell$ or $l_S$ in the argument before Eq. (7), since $l_S=\sqrt{\ell \hbar v/\Delta}$ depends monotonically on $\ell$.

I hope that the Referee now deems the manuscript suitable for publication in SciPostPhys.

Reply to "Anonymous Report 2"

I would like to thank the Referee for her/his detailed and globally positive report on my manuscript. I am also grateful for the critical questions she/he raised. They are addressed in detail below. The manuscript has been changed accordingly, and her/his comments have helped me to improve it, namely in view of its accessibility to the readers.

1.) The first question is concerned with the Faraday and Voigt geometry, which the Referee prefers to be defined with respect to the polarization plane of the photon. It is true that the wave vector is small within optical processes in matter that are essentially vertical with no wave-vector transfer due to the large speed of light as compared, e.g., to the Fermi velocity. Even if it is small in modulus, the photon nevertheless has a wave vector (and thus a direction of propagation) that is perpendicular to the polarization plane, as a consequence of Maxwell's equations, $\nabla\cdot B=\nabla\cdot E=0$, that read $q\cdot B= q\cdot E=0$ in reciprocal space. Here, E and B are the electric- and magnetic-field components of the photon, respectively, that span the polarization plane, and q is the wave vector. The two different geometries can thus and are often defined with respect to the orientation of the wave vector. A footnote has been added to clarify this point.

  1. and 2.bis) Section II provided indeed a succinct introduction to the physical properties of interface states in a topological heterojunction, based on previous studies. In view of the Referee's useful comment, I agree that it might have been too succinct and that further information is useful to render the manuscript "self-contained". The section has therefore been enhanced with the introduction of a generic model Hamiltonian and the main steps in its diagonalization in the presence of a smooth interface. This explicit presentation of the model allows hopefully for a better appreciation of the analogy between the gap variation across the interface and the "fake" magnetic field as well as for understanding how the effective width parameter may be controlled by an inplane magnetic field.

3.) The Referee mentions that the present proposal requires pumping by a source in the THz regime and that she/he does thus not see the interest in the setup as a source of THz light. The interest of the present setup, in the case of resonant pumping, is that the source does not need to be coherent while the outcoming (laser) light is expected to be coherent. I agree that, in order to truly realize the effect, several intermediate steps (such as the measurement of cyclotron emission) are required to test the proposal. However, I would like to point out that one might also possibly use the bulk conduction band as a possible reservoir of electrons for stimulated light emission. In this case, one might pump at higher frequencies. This aspect is discussed in Sec. V below Eq. (15) [former Eq. (11)].

3.bis) The Referee asks for further justification of formula (31). In my response, I understand that she/he refers to the Eq. (11), which is now Eq. (15). The expression is indeed obtained directly from Ref. [33], see e.g. Eq. (20), with the help of Fermi's golden rule. The matrix element is given in terms of the effective Coulomb interaction, which is essentially given by $e^2/\epsilon l_B$ in the lower Landau levels (apart from a numerical prefactor, and in higher Landau levels the magnetic length needs to be replaced by the cyclotron radius $R_C~l_B\sqrt{2n}$) and the Dirac function is broadened by $1/\tau$ in terms of the dephasing time $\tau$. At resonance, when the frequency $\hbar\omega$ coincides with the level spacing $\Delta E$) a Lorentzian-type distribution $\hbar/\tau/[(\hbar\omega-\Delta E)^2+\hbar^2/\tau^2]$ then becomes $\tau/\hbar$, as in Eq. (11). Some intermediate steps have been added to justify the equation, but I have chosen not to provide (known) details about the form factors arising from the wave functions. They give lengthy expressions in terms of Laguerre polynomials that are eventually reduced to a numerical prefactor and that do not affect the argument based on simple orders of magnitude. However, I agree that they may be useful (and will be given) in a more complete study of the transition rates (see below in my response to point 5.bis).

  1. and 5.) The figures have been changed in response to the Referee's suggestions. To show that there are no other resonant levels, Fig. 1 has been changed. Both, higher VP surface bands and the bulk conduction and valence bands have been added. I have chosen not to add further levels to Figs. 2 and 3 for reasons of visibility. However, the Fermi level has been added to both of them, in response to the Referee's request, and a different colour code has been adopted for the pump and emission processes (magenta and orange, respectively, not to use the same colours as for the Landau levels associated with the different VP bands).

5.bis) I agree with the Referee that a more detailed and quantitative study of the transition rates and a balance equation is required to test the present proposal. However, I most respectfully disagree with the Referee that such a study, which is beyond the scope of the present manuscript presenting the proposal, be required at this stage. The present manuscript is not meant to be a complete theoretical solution of the THz-laser problem in terms of topological heterojunctions, but it is a condensed-matter proposal that I deem promising, in line with the other Referee who states that she/he is "confident that this convincing idea will stimulate further promising studies". I therefore leave a more detailed study of the transition rates and a balance equation -- along with the necessary experimental studies -- for future work. The scope of the present manuscript and the need for further studies is now more clearly stated at the end of Sec. V.

6.) As a theoretician who is mainly concerned with fundamental research, I am not an expert in technological applications. However, there seems to be an apparent lack of coherent light sources in the THz regime (called the "THz gap") as compared to lasers in the visible and infrared range on the one-hand side and the microwave range on the other hand. Generically, a working Landau-level laser would be precisely situated in this frequency range. A sentence pointing out this technological interest has been added to the conclusions along with a new reference to a so-called "road-map".

7.) I am particularly grateful to the Referee for pointing out the length of the paragraphs, a possible fault related to my German origins, often more prominent in outrageously long sentences [Twain1880]. The revised version of the manuscript takes into account this linguistic criticism, and the paragraphs have been shortened or cut into sub-paragraphs.

I hope that the Referee now deems the revised manuscript suitable for publication in SciPostPhys.

References (not added to the bibliography of the main text)

[Twain1880] Mark Twain, "The Awful German Language", in "A Tramp Abroad" (1880); open access: https://faculty.georgetown.edu/jod/texts/twain.german.html

---

## Round 3 · List of Changes

Major changes in response to the Referees (minor changes not listed)

  • enhanced discussion of the basic model in Sec. II, containing a derivation of the spectra, a clearer discussion of the analogy between the gap variation at the interface and a fake magnetic field, a quantitative reminder of how the effective interface width can be tuned with the help of an in-plane magnetic field
  • change of the figures; Fig. 1 contains more levels as well as the bulk bands to allow the reader to appreciate the absence of degenerate transitions; Fermi level has been added to Figs. 2 and 3, and colors for the absorption/emission processes have been changed
  • a short argument (and a footnote) added to justify the choice in the definition of the Faraday/Voigt geometry with respect to the direction of the photon's wave vector, which is equivalent (via Maxwell's equations) to a definition in terms of the photon's polarization
  • further information provided about Fermi's golden rule in the calculation of rapid interaction-induced relaxation processes around Eq. (15) [former Eq. (11)]; it remains an order-of-magnitude argument
  • short discussion about the relevance of relaxation processes added to Sec. V to underline the scope of the present manuscript
  • discussion added to Sec. V about possible reabsorption processes by chiral surface at the opposite side, as well as possibility to block such processes
  • sentence added to the conclusions about the technological relevance of (coherent) THz radiation source ("THz gap")

---

## Round 4 · Author Response

Warnings issued while processing user-supplied markup:
- Inconsistency: plain/Markdown and reStructuredText syntaxes are mixed. Markdown will be used.
Add "#coerce:reST" or "#coerce:plain" as the first line of your text to force reStructuredText or no markup.
You may also contact the helpdesk if the formatting is incorrect and you are unable to edit your text.
Dear Editor,
Thank you very much for sending me the second report of Referee 2, who made her/his justified criticism about the definition of the geometry much clearer now. She/he is fully right that in the Voigt geometry, the photon's electric field must be aligned with the external magnetic field to obtain the desired optical selection rules. The main text has been changed in response to the referee's criticism, along with the other two suggestions.
Sincerely yours, Mark Oliver Goerbig
Response to the Referee's second report:
I would like to thank the Referee for her/his second report and the helpful clarification of the criticism mentioned in her/his first report. I now understand her/his point better and fully agree that, in the Voigt geometry, also the orientation of the photon's electric field must be specified (in contrast to the more commonly used Faraday geometry, where the direction of the photon's propagation is sufficient). Indeed, in order to obtain the selection rule n->n in Landau-level spectroscopy, one must ensure that the photon's electric field be in the same direction as the external magnetic field. Otherwise, one retrieves the n-> n+-1 selection rules as in the Faraday geometry, which does not have this ambiguity.
In response to the Referee's comment, I have now added this precision of the direction of the photon's electric field in the discussion of the Faraday/Voigt geometry both in the main text as well as in the added footnote [32]. Furthermore, I also mention the polarization in the caption of Table 1.
Furthermore, I have taken into account the Referee's suggestion to make clear the circular polarizations $\sigma^+$ and $\sigma^-$ of the photon in the coupling to the LL transitions in the Faraday geometry.
Finally, I repeat in the caption of Figs. 2 and 3 that the dashed lines correspond to the dispersion of the surface bands in the absence of a magnetic field, as in Fig. 1.

---

## Round 4 · List of Changes

- clarification of the orientation of the photon's electric field with respect to the external magnetic field in the Voigt geometry (a) in the main text, (b) in the footnote [32], and (c) in the caption of Table I.
- coupling to the two circular polarizations (for the Faraday geometry) added to the main text (last paragraph of Sec. II)
- meaning of the dashed lines in Figs. 2 and 3 specified in the caption (dispersion in the absence of a magnetic field)

---

## Editorial Decision

published